# ACE2-Independent Interaction of SARS-CoV-2 Spike Protein with Human Epithelial Cells Is Inhibited by Unfractionated Heparin

**DOI:** 10.3390/cells10061419

**Published:** 2021-06-07

**Authors:** Lynda J. Partridge, Lucy Urwin, Martin J. H. Nicklin, David C. James, Luke R. Green, Peter N. Monk

**Affiliations:** 1Department of Molecular Biology and Biotechnology, The University of Sheffield, Sheffield S10 2TN, UK; l.partridge@sheffield.ac.uk; 2Department of Infection, Immunity and Cardiovascular Disease, The University of Sheffield, Sheffield S10 2RX, UK; lurwin2@sheffield.ac.uk (L.U.); m.nicklin@sheffield.ac.uk (M.J.H.N.); l.r.green@sheffield.ac.uk (L.R.G.); 3Department of Chemical and Biological Engineering, The University of Sheffield, Sheffield S1 4NL, UK; d.c.james@sheffield.ac.uk

**Keywords:** SARS-CoV-2, heparin, glycosaminoglycan, spike protein

## Abstract

Coronaviruses such as SARS-CoV-2, which is responsible for COVID-19, depend on virus spike protein binding to host cell receptors to cause infection. The SARS-CoV-2 spike protein binds primarily to ACE2 on target cells and is then processed by membrane proteases, including TMPRSS2, leading to viral internalisation or fusion with the plasma membrane. It has been suggested, however, that receptors other than ACE2 may be involved in virus binding. We have investigated the interactions of recombinant versions of the spike protein with human epithelial cell lines that express low/very low levels of ACE2 and TMPRSS2 in a proxy assay for interaction with host cells. A tagged form of the spike protein containing the S1 and S2 regions bound in a temperature-dependent manner to all cell lines, whereas the S1 region alone and the receptor-binding domain (RBD) interacted only weakly. Spike protein associated with cells independently of ACE2 and TMPRSS2, while RBD required the presence of high levels of ACE2 for interaction. As the spike protein has previously been shown to bind heparin, a soluble glycosaminoglycan, we tested the effects of various heparins on ACE2-independent spike protein interaction with cells. Unfractionated heparin inhibited spike protein interaction with an IC_50_ value of <0.05 U/mL, whereas two low-molecular-weight heparins were less effective. A mutant form of the spike protein, lacking the arginine-rich putative furin cleavage site, interacted only weakly with cells and had a lower affinity for unfractionated and low-molecular-weight heparin than the wild-type spike protein. This suggests that the furin cleavage site might also be a heparin-binding site and potentially important for interactions with host cells. The glycosaminoglycans heparan sulphate and dermatan sulphate, but not chondroitin sulphate, also inhibited the binding of spike protein, indicating that it might bind to one or both of these glycosaminoglycans on the surface of target cells.

## 1. Introduction

SARS-CoV-2, the causative agent of COVID-19, is thought to infect cells after binding with high affinity to a host cell receptor, ACE2 [1]. The ACE2-binding domain is located in the spike protein that consists of two regions: S1, which includes the high-affinity receptor-binding domain (RBD) and S2, containing sequences necessary for fusion with the host cell. S1 and S2 are linked by a sequence that has a putative furin cleavage site that promotes the infection of human cells [2]. A host cell surface serine protease, TMPRSS2, is also proposed to be involved in viral entry by cleaving S1 and S2, leading to activation of the fusion machinery [1]. By analogy with SARS-CoV-1, it is expected that the virus can fuse with the plasma membrane or with endosomal membranes following internalisation (reviewed in [3]).

Paradoxically, ACE2 is expressed at quite low levels by most cell types [4], leading to suggestions that additional receptor sites may exist. One such additional site has recently been identified as neuropilin-1 (NRP-1), a growth factor co-receptor [5]. Viruses, such as herpes simplex and the β coronavirus family, are known to interact with host glycosaminoglycans (GAG) [6,7]. A growing body of evidence suggests that SARS-CoV-2 can bind GAGs such as heparan sulphate and heparin, dependent on their level of sulphation [6,8,9,10] and that heparin can inhibit SARS-CoV-2 entry into host cells [11,12,13]. Initial binding to heparan sulphates is thought to keep the spike protein in an ‘open’ conformation, allowing for downstream binding and processing by ACE2 and TMPRSS2, respectively [8].

Here, we show binding of SARS-CoV-2 spike protein to human epithelial cell lines that differentially express ACE2 and TMPRSS2. The intact viral spike protein, but not the isolated S1 or RBD subunits, exhibits a temperature-dependent cellular interaction that allows rapid detection by flow cytometry. We have used this assay to confirm that heparin can inhibit spike protein interactions with human cells and demonstrate that high-affinity heparin binding might involve the putative furin cleavage site and the S2 region of the viral spike protein but that it is largely independent of the S1 region.

## 2. Materials and Methods

### 2.1. Materials

Unfractionated heparin (Leo, 1000 U/mL), dalteparin (25,000 IU/mL) and enoxaparin (10,000 IU/mL) were obtained from the Royal Hallamshire Hospital Pharmacy, Sheffield, UK. Fondaparinux was purchased from Merck, Gillingham, UK. Goat anti-human ACE2 antibody AF933 (Biotechne, Abingdon, UK), goat control IgG AB-108-C (Biotechne, Abingdon, UK), rabbit anti-human TMPRSS2 (MBS9215011, Gentaur, Potters bar, UK), and rabbit IgG control (Biolegend, San Diego, CA, USA) were used as per manufacturers’ instructions. Protamine sulphate was obtained from Royal Hallamshire Hospital Pharmacy, Sheffield, UK, surfen hydrate from Merck, Gillingham, UK, camostat mesylate from Biotechne, Abingdon, UK, GM6001 from Sigma, UK and glycosaminoglycans from Iduron, Alderley Edge, UK. His6-tagged human complement fragment 5a des Arg was obtained from Hycult Biotech, Uden, NL.

### 2.2. Cell Culture

The A549 lung carcinoma cell line (European Collection of Animal Cell Cultures Salisbury, UK), the human keratinocyte cell line, HaCaT (Cell Line Services GmbH, Eppelheim, DE), the VERO E6/TMPRSS2 cell line (National Institute for Biological Standards Potters Bar, UK), and the HEK293 cell line (American Tissue Type Collection (ATCC) Manassas, VA, USA) were all routinely cultured in DMEM with 10% foetal calf serum (FCS). ACE2-transfected ACE2HEK293 cells were kindly provided by Paul Bieniasz (The Rockefeller University, USA), cultured as described for wild-type (wt)HEK293 cells including selection with 5 ug/mL blasticidin. The RT4 human bladder cell line (ATCC, Manassas, VA, USA) was cultured in McCoys 5A medium (Fisher Scientific, Loughborough, UK) supplemented with 10% FCS. HCE2 corneal epithelial cell line (ATCC, Manassas, VA, USA) was cultured in keratinocyte serum-free medium supplemented with bovine pituitary extract, insulin, hydrocortisone and epidermal growth factor. The Caco2 colorectal adenocarcinoma cell line was a gift from Dr. Michael Trikic (University of Sheffield, UK) and cultured in EMEM supplemented with 10% FCS plus non-essential amino acids. Cell lines were routinely sub-cultured by trypsinisation and maintained in sub-confluent cultures.

### 2.3. Real Time Quantitative PCR (RT-qPCR)

HaCaT, RT4, HCE2, A549, wtHEK293, ACE2HEK293 and Caco2 cell lines were cultured for 48 h and harvested using trypsin/EDTA. Total RNA was extracted using the RNeasy Mini Kit (Qiagen, Manchester, UK) and quantified using a NanoPhotometer N60 Touch (Cole-Palmer UK, St Neots, UK). RNA samples were converted to cDNA using the High-Capacity cDNA Reverse Transcription Kit (Fisher Scientific, Loughborough, UK). Control samples containing no reverse transcriptase or no RNA template were included. RT-qPCR was performed by SYBR Green assay using the PrecisionPLUS OneStep RT-qPCR Master Mix (Primer Design, Chandler’s Ford, UK) and the QuantStudio 5 Real-Time PCR system (Fisher Scientific, Loughborough, UK). Gene expression levels of ACE2 and TMPRSS2 were investigated by comparative CT experiments and GAPDH expression was measured as an endogenous control. All primers were designed using PrimerBLAST (see Appendix A for primer sequences). The wtHEK293 cell line has been used as a reference for ACE2/TMPRSS2 gene expression and the data are presented as the mean relative quantification (ΔΔCq).

### 2.4. Coronavirus Spike Proteins

Spike protein binding to cells was tested using four versions of the SARS-CoV-2 spike protein and one version of the SARS-CoV-1 spike protein. Wild-type SARS-CoV-1 spike protein (S1S2; Tor2 isolate M1-P1195, S577A) and SARS-CoV-2 spike protein (wtS1S2; Val16-Pro1213; Stratech, Ely, UK) with His6 tags at the C-terminus were expressed in baculovirus-insect cells, while S1-Fc (V16-R685; Stratech, Ely, UK) with a mouse IgG1 Fc region at the C-terminus was expressed in HEK293 cells. Mutant SARS-CoV-2 spike protein (mS1S2) and the receptor-binding domain (RBD) cloned into a pCAGGS expression vector were kindly provided by Florian Krammer (Mount Sinai, USA) [10]. For mS1S2, a polybasic cleavage site, recognised by furin, was removed (^682^RRAR^685^ to A) in mS1S2 and two stabilising mutations were added (K986P and V987P). A thrombin cleavage site, a T4 foldon sequence allowing trimerisation and a His6 tag were fused to the C-terminal amino acid P1213. RBD was expressed using the natural signal peptide fused to RBD (R319-Q541) and a His6 tag at the C terminus. Recombinant proteins were expressed and purified as described previously [11].

### 2.5. Spike Protein Interaction Assay

Cells were harvested by brief trypsinisation and added to wells of a 96-well U-bottom plate. After centrifugation at 300× *g* for 2 min and washing with HBSS containing divalent cations and 0.1% BSA (assay buffer, AB), cells were incubated with potential inhibitors in AB for 30 min at 37 °C. The supernatant was removed following centrifugation and AB containing spike protein added before incubation at 4 or 37 °C for 60 min. Cells were washed once and then incubated with the appropriate fluorescently labelled secondary antibody (anti-mouse polyvalent Ig-FITC (Merck, Gillingham, UK) or anti-His6 HIS.H8 DyLight 488, (Fisher Scientific, Loughborough, UK) for 30 min at room temperature. Cells were finally resuspended in AB containing propidium iodide and cell-associated fluorescence measured using a flow cytometer. Live cells were gated as a propidium iodide negative population and the median fluorescence (MFI) of at least 3000 cells recorded. MFI was calculated after subtraction of cell-associated fluorescence of the secondary antibody alone.

### 2.6. Determination of Spike Protein Binding to Heparin by ELISA

Heparin-binding plates (Plasso EpranEx™), a gift from Dr David Buttle (University of Sheffield, UK), were coated with 10 µg/mL of UFH or dalteparin diluted in phosphate buffered saline (PBS) at room temperature overnight. Wells were washed twice with PBS 0.05% Tween (PBST), blocked with PBST with 1% BSA for 2 h at 37 °C before three further washes with PBST. Various concentrations of His-tagged spike proteins in AB buffer (or AB buffer control) were added to the wells and incubated at 37 °C for 2 h. For competition assays, proteoglycans were added at the same time as the spike proteins. Wells were washed three times as above then incubated at room temperature with biotin-labelled rabbit monoclonal anti-His6 (Fisher Scientific, Loughborough, UK) diluted to 1/1000 in AB for 1 h, washed 3 times with PBST and incubated for 30 min with streptavidin-HRP (Fisher Scientific, Loughborough, UK) diluted 1/200 in AB. After washing 3 times with PBST and twice with dH_2_O, TMB substrate solution (Fisher Scientific, Loughborough, UK) was added followed by 1M HCl to quench the reaction. Absorbance was measured at OD_450_ nm with non-specific spike protein binding to uncoated wells subtracted.

## 3. Results

### 3.1. ACE2 and TMPRSS2 mRNA Expression within Human Cell Lines

We investigated the binding of SARS spike proteins to human epithelial cell lines with a range of low levels of ACE2 and TMPRSS2 expression. The mRNA for ACE2 and TMPRSS2 in a variety of human cell lines was measured by quantitative PCR and compared to levels in the commonly used human embryonic kidney 293 cell line (HEK293), reported to have very low ACE2 mRNA levels [14,15]. ACE2-transfected HEK293 cells expressed 357-fold higher ACE2 than wild-type (wt)HEK293 but had similar TMPRSS2 levels (Figure 1). HaCaT skin keratinocytes had the highest native ACE2 mRNA expression (2.3-fold higher than wtHEK293) but there was very low expression of TMPRSS2 (Figure 1). HCE-2 corneal epithelial cells were also investigated due to the proposed role of the ocular surface as a route of infection [16] but little or no ACE2 or TMPRSS2 expression was detected. The urinary bladder epithelial cell line, RT4 [15], had ACE2 and TMPRSS2 expression 0.5-fold lower and 52-fold higher, respectively, than in wtHEK293 cells. In contrast, the Caco2 colorectal adenocarcinoma cell line, used in several infection studies of SARS-CoV and SARS-CoV-2 [17], has 0.47-fold lower and 56-fold higher mRNA for ACE2 and TMPRSS2, respectively, compared to wtHEK293 cells. Finally, the human lung adenocarcinoma alveolar basal epithelial cell line A549 expresses very low levels of both ACE2 and TMPRSS2, perhaps explaining why this cell line does not support infection by SARS-CoV-2 viral particles [17]. As RT4 cells seemed to have low ACE2 and TMPRSS2 expression levels representative of other epithelial cell types and are also known to express ADAM17 [14,15], a metalloprotease known to be involved in the processing of ACE2 [18], these cells were chosen to investigate possible ACE2-independent binding to epithelial cells.

### 3.2. SARS-CoV-2 Spike Protein Interaction with Human Cells Is Temperature Dependent

To detect spike protein interactions with cells, we used recombinant S1 tagged with mouse Fc; His6-tagged RBD or intact wtS1S2 proteins, and the appropriate fluorescently labelled secondary antibodies to stain RT4 cells for flow cytometry. At 4 °C, only a very low level of wtS1S2 binding was detected, while S1 and RBD binding was undetectable (Figure 2), with similar results at 21 °C (data not shown). Surprisingly, interaction of wtS1S2 at 37 °C was much stronger than at 4 °C, whereas S1 and RBD protein association was still very low (Figure 2). A His6-tagged human complement fragment 5a des Arg was used to test the specificity of the wtS1S2 interaction at 330 nM; no binding was detected at either 4 or 37 °C (data not shown). The temperature dependency of the cellular interaction with wtS1S2 suggests that the protein might undergo a conformational change, perhaps as a result of proteolytic processing at the cell surface, which allows binding to occur. However, neither the TMPRSS2 inhibitor camostat mesylate nor the ADAM17 inhibitor GM6001 had any effect on wtS1S2 binding at concentrations up to 100 µM (data not shown).

### 3.3. wtS1S2 Can Interact with a Range of Human Cells Independent of ACE2 Expression

We next investigated the interaction of wtS1S2 to epithelial cells with varying degrees of ACE2 expression at 37 °C. A 100 nM wtS1S2 was able to attach to all cells tested, with higher ACE2-expressing cells having only slightly higher levels of interaction (relative to background RT4-1.80 ± 0.32; Caco2-1.75 ± 0.40; HaCaT-1.66 ± 0.16; Figure 3A). Despite high mRNA expression by ACE2-transfected HEK293 cells, attachment of wtS1S2 is comparable between wtHEK293 cells (1.44 ± 0.27) and ACE2HEK293T cells (1.60 ± 0.28). A549 cells, which have no detectable levels of ACE2, were also able to associate with wtS1S2 (1.33 ± 0.20), suggesting a largely ACE2-independent interaction with all of these cell types.

As we observed wtS1S2 interaction with cells expressing little or no ACE2, we sought to further investigate the role of spike protein interaction with ACE2 by assessing the concentration dependence of wtS1S2 and RBD attachment to wtHEK293 and ACE2HEK293 cells. wtS1S2 and RBD interacted with 293T_ACE2_ cells with similar affinities (EC_50_ = 137.7 nM and 49.1 nM, respectively) (Figure 3B). In contrast, wtS1S2 also associated with wtHEK293 cells expressing very low levels of endogenous ACE2, albeit at a slightly lower affinity (EC_50_ = 784.6 nM), while the interaction of RBD with these cells was not above background. This confirms that a major component of the interaction of wtS1S2 with cells is independent of ACE2, perhaps via an alternative receptor, whereas interaction of the RBD alone requires high levels of cell surface ACE2.

The dependence of RBD binding on ACE2 was confirmed using VERO E6/TPMRSS2 cells that express levels of endogenous African Green Monkey ACE2 high enough to support SARS-CoV-2 entry [19]. These cells bound significant levels of RBD at 1000 nM, whereas binding to RT4 was much lower (Figure 4). Due to the possibility that the treatment of cells with trypsin might interfere with RBD binding, A549 cells, harvested with either trypsin/EDTA (TE) or non-enzymatic cell dissociation solution (CDS) were tested. RBD binding was undetectable under both conditions (Figure 4), confirming the role of ACE2 in the binding of RBD.

### 3.4. wtS1S2 Interactions May Require a Putative Furin-Binding Site

The polybasic site between the S1 and S2 regions (^681^PRRARSV^687^) is a putative furin cleavage site that promotes the infection of human cells [2] but is not essential [20]. The spike protein from SARS-CoV-1 does not contain this site [21]. We compared the binding of wtS1S2 from SARS-CoV-1 and SARS-CoV-2 to RT4, A549 and VERO E6/TMPRSS2 cells. At all concentrations tested, SARS-CoV-1 wtS1S2 bound to these cells more weakly than SARS-CoV-2 wtS1S2 (Figure 5). Interestingly, harvesting A549 cells with CDS rather than trypsin increased the binding of both wtS1S2 proteins (Figure 5), in contrast to RBD binding, which was still undetectable (Figure 4).

A mutant spike protein (mS1S2) protein lacking the putative furin cleavage site also demonstrated dramatically reduced interaction with RT4 cells compared to wtS1S2, similar to the S1 RBD alone (Figure 6A,B). This might be explained by the lack of proteolytic processing of the spike protein during viral association with the cell, although we cannot rule out the importance of a pair of stabilising modifications, 300 amino acids C-terminal of the polybasic cleavage. We were unable to determine the affinity of the interaction due to limited availability of recombinant wtS1S2 but association was still increasing even at 330 nM (Figure 6A), suggesting the limiting step is a relatively low-affinity interaction. This is in contrast to the affinity of RBD or S1 for ACE2HEK293 cells shown here (Figure 3) and in previously published reports [18].

### 3.5. Unfractionated Heparin Inhibits wtS1S2 Binding to RT4 Cells

Having developed an assay that mimics some aspects of the interaction of SARS-CoV-2 with epithelial cells, we used it to test potential inhibitors. Although an anti-ACE2 antibody has previously been reported to block viral binding to host cells [1], preincubating RT4 cells with the same antibody caused a small but significant increase in wtS1S2 interaction (54.25% above untreated cells, *p* = 0.0025) (Appendix A). Antibody-mediated cross-linking of the low levels of ACE2 at the cell surface may allow more rapid attachment of the spike protein due to receptor clustering. Heparin has been reported to bind directly to S1 and to interfere with SARS-CoV-2 infection [7] and so we tested the effects of pre-incubating RT4 cells with heparin on the wtS1S2 attachment at 37 °C. Unfractionated heparin (UFH) at 10 U/mL inhibited 80% of 330 nM wtS1S2 interaction with the cells (Appendix A) and was significantly reduced compared to untreated controls.

Using 100 nM wtS1S2, the inhibition by UFH was complete with an IC_50_ of 0.033 U/mL (95% confidence interval 0.016–0.07) (Figure 7). This is far below the target prophylactic and therapeutic concentrations of UFH in serum, 0.1–0.4 U/mL and 0.3–0.7 U/mL, respectively [19,20]. In contrast, two low-molecular-weight heparins (LMWH), dalteparin and enoxaparin, only gave partial inhibition, and were less potent than UFH (IC_50_ values of 0.558 and 0.072 U/mL, respectively). Typical prophylactic and therapeutic serum concentrations of LMWH are 0.2–0.5 U/mL and 0.5–1.2 U/mL [21], respectively, suggesting that dalteparin used prophylactically would be below the effective dose required for inhibition of viral infection. The synthetic pentasaccharide heparinoid, fondaparinux, had no effect on S1S2 interaction at concentrations up to 0.1 mg/mL, although its therapeutic concentration is <2 µg/mL [22] (Figure 7).

Protamine sulphate is a highly cationic peptide that is used clinically to reverse anti-coagulant activity [22] by dissociating heparin-antithrombin III complexes. Surfen is a GAG-binding molecule that binds to heparin and also to the GAGs, dermatan (DS), chondroitin (CS) and heparan sulphates (HS) [23]. Both of these compounds inhibited wtS1S2 binding to RT4 cells in a dose-dependent manner, with IC_50_ values of 1.7 µg/mL and 0.14 µg/mL for surfen and protamine, respectively (Figure 8). Interestingly, at higher concentrations, both agents induced increased levels of wtS1S2 binding reaching control values in the case of surfen, reached control values at 250 µg/mL (Figure 8). These data indicate that cell surface proteoglycans such as HS may act as SARS-CoV-2-binding sites on host cells. The reason for the increased binding of wtS1S2 at higher concentrations of protamine and surfen is currently unclear.

### 3.6. wtS1S2 Lacking the Furin Cleavage Site Has a Lower Affinity for Heparin Than Wild-Type S1S2

The polybasic site furin cleavage site in wtS1S2 is also suggested to be a heparin-binding site [23], so we inferred that altered heparin binding by the mS1S2 might be linked to its reduced association with RT4 cells. In ELISA, mS1S2 and RBD had a significantly lower affinity for both UFH (mS1S2, EC_50_ = 217.8 nM; RBD, EC_50_ = 818.4 nM) and LMWH (mS1S2, EC_50_ = 162.2 nM; RBD, EC_50_ = 288.3 nM) relative to wtS1S2 (EC_50_ = 6.8 nM and 9.3 nM, respectively) (Figure 9). It is likely that wtS1S2 contains multiple binding sites for heparin that confer high avidity including the polybasic furin cleavage site. Additional sites may be present in S1 and RBD but these are of relatively lower affinity. Interestingly, wtS1S2 bound to both UFH and dalteparin with similar affinities and so binding affinity for immobilised heparins does not obviously explain the difference between UFH and LMWH in the inhibition of cell binding.

### 3.7. Effects of Heparinases and Trypsinisation on wtS1S2 Binding

If HS acts as a major binding site for wtS1S2 on epithelial cells, wtS1S2 binding should be inhibited by enzymatic cleavage of HS from the cell surface. Trypsin is known to rapidly cleave membrane proteins that display GAGs [24] and wtS1S2 binding to A549 cells was increased by cell harvesting in the absence of trypsin (Figure 5). We used different methods of cell harvesting to investigate whether this could affect wtS1S2 binding to RT4 cells. These cells were also treated with a mixture of heparinase I and III, known to cleave the majority of HS and then harvested using CDS or TE. Harvesting by CDS caused a small, non-significant increase in binding to RT4 cells relative to TE (Figure 10A). Heparinase treatment of trypsinised cells caused only a small, non-significant reduction in wtS1S2 binding (Figure 10A) but there was a significant decrease following heparinase treatment of non-enzymatically harvested cells (Figure 10A). Antibody 3G10, which recognises a neo-epitope produced by heparinase cleavage of HS, bound strongly to non-enzymatically harvested, heparinase-treated cells but only weakly to trypsinised cells (Figure 10B). This indicates that trypsin can cleave membrane proteins that bear HS, removing most but not all of the cell surface HS. wtS1S2 may still be able to bind to this small amount of remaining HS and so binding is relatively insensitive to trypsin and heparinase treatment. Alternatively, wtS1S2 may bind to GAGs other than HS. However, chondroitinase treatment of RT4 cells prior to trypsinisation did not result in a loss of wtS1S2 binding (100.5 ± 2.5% of the untreated control).

### 3.8. Inhibition of wtS1S2 Binding by Different GAGs and Heparins

To investigate the relative effects of different GAGs and heparin analogues on binding, we screened a large panel of heparin molecules with different chain lengths (4–16 dipolysaccharide units), molecules with normal sulphation or lacking 2-O, 6-O or N-sulphation (2-O, 6-O and N, respectively), and different GAGs such as HS, CS and DS. HS inhibited wtS1S2 binding to RT4 cells strongly at 100 µg/mL, whereas DS caused moderate inhibition and CS no inhibition at all. The chain length was critically important for the heparins, with shorter, fully sulphated heparin failing to inhibit binding (Figure 11) and only moderate inhibition by longer chain (dp12–16) heparins. Heparins lacking sulphates at 2-0, 6-O and N sites were all unable to inhibit wtS1S2 binding (Figure 11).

We also investigated whether GAGs inhibited binding of wtS1S2 and RBD to immobilised UFH in ELISA. For wtS1S2 (Figure 12), HS and DS both significantly inhibit binding to UFH (*p* = 0.0045 and 0.006, respectively), whereas the RBD binding was not significantly inhibited even by HS (Figure 12). The shorter-chain-length heparins did not significantly inhibit the binding of either protein to UFH (Figure 12), except for dp10, which caused moderate but significant inhibition of wtS1S2 binding (*p* = 0.027). Thus, it is possible that S1S2 but not RBD can bind to HS and to other GAGs at the cell surface such as DS.

## 4. Discussion

Using a flow cytometric assay, we have demonstrated that intact recombinant wtS1S2 spike protein associated with multiple human cell lines independently of ACE2 expression, while the S1 or RBD region alone did not. However, SARS-CoV-2 RBD interacts strongly with cell lines when ACE2 is expressed at high levels. Using this assay, we have further shown that UFH and two low-molecular-weight heparins (LMWH) in clinical use inhibit ACE2-independent wtS1S2 binding; these treatments have previously been shown to reduce binding of both pseudovirus containing SARS-CoV-2 spike protein and SARS-CoV-2 virus [11,12,13].

Whilst other studies have used African green monkey VERO E6 cells or human cells overexpressing relevant SARS-CoV-2 receptors, we have focused on native human cell lines. Although we demonstrate that ACE2 is involved in RBD binding, the intact spike protein interacted with cells in the absence of detectable ACE2, suggesting the presence of additional receptors. The ability of the SARS-CoV-2 spike protein to attach to various GAGs provides a host of candidate proteins at the cell surface [6,7,8,9]. In support of this, studies have suggested the importance of differing glycan sulphation states in different tissues as an explanation for viral tropism. Recently, SARS-CoV-2 spike protein S1 has been shown to bind HS with varying degrees of sulphation with differing affinities; chain length and 6-O-sulphation were shown to be particularly important [8].

In contrast to a previous publication [25], we did not observe binding of RBD to A549 cells lacking ACE2. It is unclear why our ACE2-binding RBD did not interact with GAGs on A549 cells but this may be due to differences in production methods.

We demonstrate that UFH and, to a lesser extent, LMWH are potent inhibitors of S1S2 interaction with human cells. Heparin has been demonstrated to interact with recombinant S1 RBD and cause conformational changes, leading to the suggestion that SARS-CoV-2 might interact with host HS through the RBD during infection [7]. Our data significantly extend this observation, suggesting the presence of more than one heparin-binding site in the intact spike protein, with one at the furin cleavage site and one other in the RBD although our data do not indicate an independent HS-binding site in the RBD. As previously noted, the presence of multiple polybasic sites within the spike protein will result in a higher avidity of binding to heparin and heparan sulphates [9]. Interestingly, all the spike proteins tested had similar affinities for UFH and LMWH and so the differing abilities of these forms of heparin to inhibit S1S2 interactions with cells is not simply due to divergent affinities. However, we cannot rule out the possibility of differing spike protein glycosylation states affecting attachment of the recombinant proteins as their expression conditions varied.

Fondaparinux contains only the antithrombin III interaction site and cannot form a ternary complex with thrombin, unlike UFH [22]. This suggests that efficient inhibition of S1S2 binding may require an interaction between heparin and two different sites on the spike protein. For example, SARS-CoV-2 spike protein optimally binds hexa- and octasaccharides composed of IdoA2S-GlcNS6S, a motif abundantly present within heparin but not heparan sulphates [9]. The LMWH may contain sufficient long polymer chains to make this dual interaction but with lower efficacy than UFH. Finally, heparin could also be inhibiting host proteases, such as Factor Xa, necessary to process the spike protein [26]. However, our data demonstrated no activity of fondaparinux and lower efficacy of dalteparin despite the inhibitory nature of the two compounds on Factor Xa [12]. Fully sulphated long-chain-length heparins are required to inhibit wtS1S2 interactions with cells, again suggesting that multiple interactions between spike proteins and heparins is required for optimal inhibitory activity. Agents that can bind GAGs, surfen and protamine sulphate, can inhibit wtS1S2 binding at low concentrations and promote binding at higher concentrations. Inhibition might occur due to blocking interactions with cell surface GAGs, but higher concentrations might lead to clustering of proteoglycans, leading to increases in binding. Paradoxically, removal of cell surface GAGs by either trypsin or heparinase treatment does not fully inhibit wtS1S2 binding to cells. This suggests that GAGs other than HS, such as DS, may also be involved. Alternatively, trypsinisation might expose new binding sites; trypsin has previously been observed to promote the entry of zoonotic coronaviruses and SARS-CoV-1 into human cells, although the mechanism has not been elucidated [27,28].

A role for the putative SARS-CoV-2 receptor, NRP-1, which is reported to bind to the furin motif in the spike protein [29], cannot be ruled out although the RT4 cells used in this study do not appear to express NRP-1 [14]. NRP-1 can be modified by GAGs [30], indicating a possible link with the GAG dependence observed here. Finally, SARS-CoV-2 spike protein has been observed to cause activation of intracellular signalling pathways in lung vascular cells [31] and our data suggest that this might be a consequence of wtS1S2 binding to cell surface proteoglycans.

## 5. Conclusions

Using a simple binding assay, we have characterised a temperature-dependent, ACE2-independent binding of a recombinant form of SARS-CoV-2 S1S2 spike protein with a range of human cells. A limitation of our study is that we have not explored the mechanism of this temperature dependency, which might be different to conventional ligand/receptor or antigen/antibody binding. Further experimentation would be required to elucidate the mechanism and to rule out the possibility that longer incubations at lower temperatures would increase S1S2 association with cells. Binding is completely inhibited by heparin but appears to be only partly dependent on GAGs such as HS. This binding is distinct from previously observed GAG-dependent binding of the spike protein RBD and S1 domains and may involve the polybasic furin cleavage site located between the S1 and S2 domains. S1S2 binding to different human cell types may be linked to the actions of SARS-CoV-2 on many different organs.

## Figures and Tables

**Figure 1 cells-10-01419-f001:**
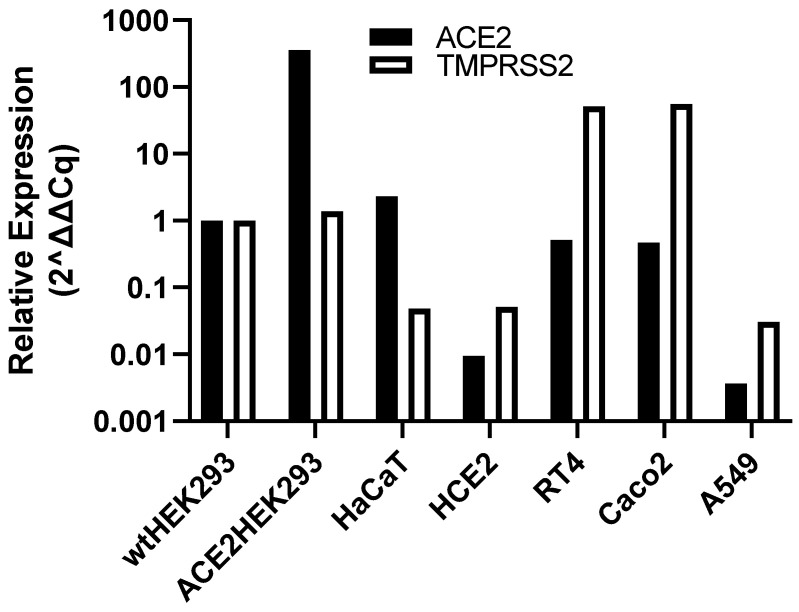
ACE2 and TMPRSS2 mRNA expression in human epithelial cell lines. Total RNA was isolated from the cell lines, converted to cDNA and ACE2 and TMPRSS2 mRNA levels determined by RT-qPCR. The data are shown relative to expression in wild-type (wt) HEK293 cells (=1) and are the means from at least 2 independent experiments.

**Figure 2 cells-10-01419-f002:**
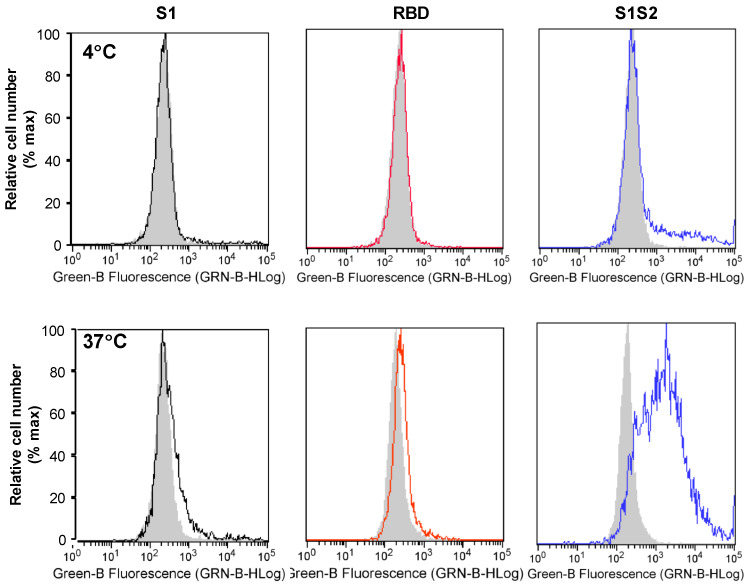
S1S2 interaction with human cells is temperature dependent. Attachment of S1, RBD and wtS1S2 (black, red and blue lines, respectively) compared to secondary-only control (solid grey), at 4 °C (Top panels) or 37 °C (Lower panels). RT4 cells were incubated with 330 nM S1-Fc, 10 µM RBD or 330 nM S1S2-His6 protein for 60 min at either 4 or 37 °C, before staining with anti-mouse Ig labelled with FITC or anti-His6 secondary antibody labelled with Dylight 488 for 30 min at 21 °C. Cell-associated fluorescence was measured by flow cytometry. Results are representative of at least 3 separate experiments.

**Figure 3 cells-10-01419-f003:**
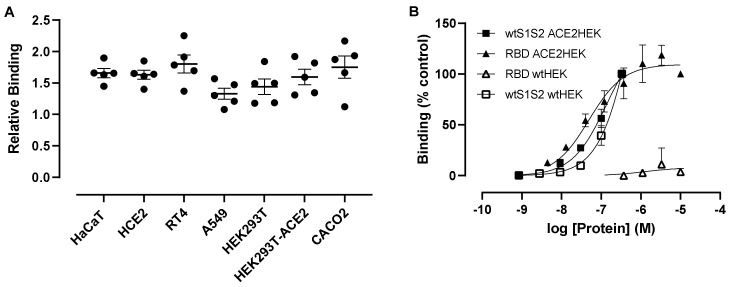
ACE2 is not required for wtS1S2 attachment to cells. (**A**) A549, wtHEK293, ACE2HEK293, Caco2, HaCaT, HCE2 and RT4 cells were incubated with 100 nM wtS1S2 for 60 min, before staining with anti-His6 secondary antibody labelled with Dylight 488. Binding of S1S2 attachment to various human cells. Data are calculated relative to the median fluorescence of secondary antibody alone (=1), means ± SEM from 5 separate experiments performed in duplicate. (**B**) Attachment of different concentrations of wtS1S2 (squares) or RBD (triangles) to wtHEK cells (open symbols) or ACE2HEK cells (filled symbols). Data are shown as a percentage of 10 µM RBD binding to ACE2HEK cells, means ± SEM from 3 independent experiments performed in duplicate.

**Figure 4 cells-10-01419-f004:**
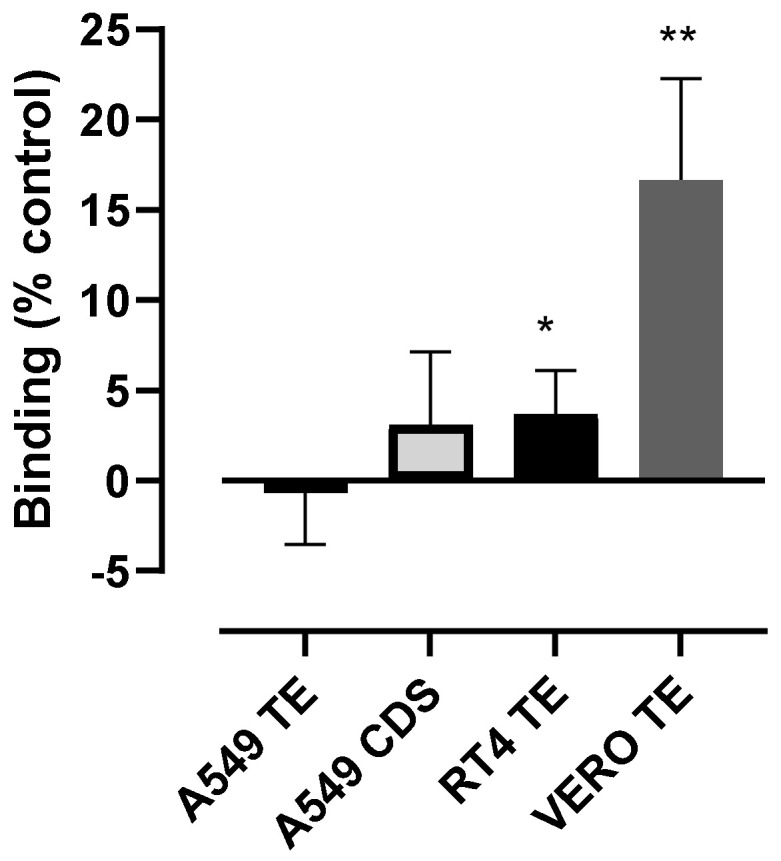
RBD can bind to VERO E6 cells that express high levels of endogenous ACE2 but not to RT4 cells. VERO E6/TMPRSS2, A549 and RT4 cells were harvested by either trypsin/EDTA or non-enzymatic cell dissociation solution (CDS) and then incubated with 1000 nM RBD for 60 min, before staining with anti-His6 secondary antibody labelled with Dylight 488. The data are shown as a percentage of the binding of 100 nM wtS1S2 and are the means ± SEM from 5–6 independent experiments performed in duplicate. The significance of the difference from 0 was assessed by a one-sample *t* test; * *p* < 0.05; ** *p* < 0.001.

**Figure 5 cells-10-01419-f005:**
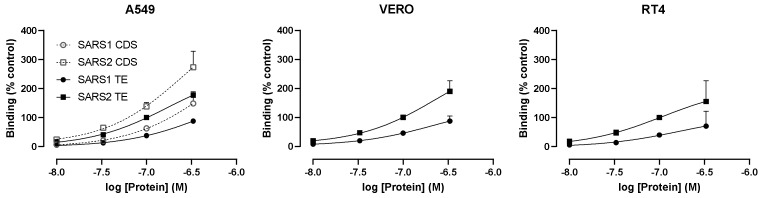
wtS1S2 from SARS-CoV-1 binds more weakly to cells than SARS-CoV-2 wtS1S2. A549, RT4 and VERO E6/TMPRSS2 cells were harvested by either trypsin/EDTA (TE; filled symbols) or non-enzymatic cell dissociation solution (CDS; open symbols) and then incubated with the stated concentrations of wtS1S2 from SARS-CoV-1 (squares) or SARS-CoV-2 (circles) for 60 min, before staining with anti-His6 secondary antibody labelled with Dylight 488. The data are shown relative to 100 nM SARS-CoV-2 wtS1S2 and are the means ± SEM from 3–4 independent experiments performed in duplicate.

**Figure 6 cells-10-01419-f006:**
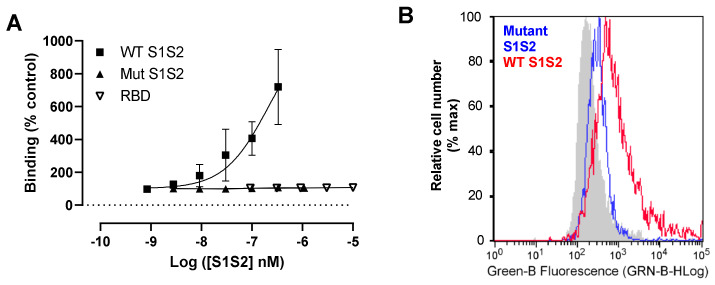
S1S2 interaction may require the furin cleavage site. (**A**) Dose–response curve for wtS1S2, mS1S2 or RBD to RT4 cells. RT4 cells were incubated with wtS1S2, mS1S2 or RBD at the stated concentrations for 60 min at 37 °C, before staining with anti-His6 secondary antibody labelled with Dylight 488. The data are the means ± SEM of 3 - 6 independent experiments performed in duplicate. (**B**) Representative histogram of 100 nM wild-type S1S2 (wtS1S2) or mutant S1S2) attachment to RT4 cells.

**Figure 7 cells-10-01419-f007:**
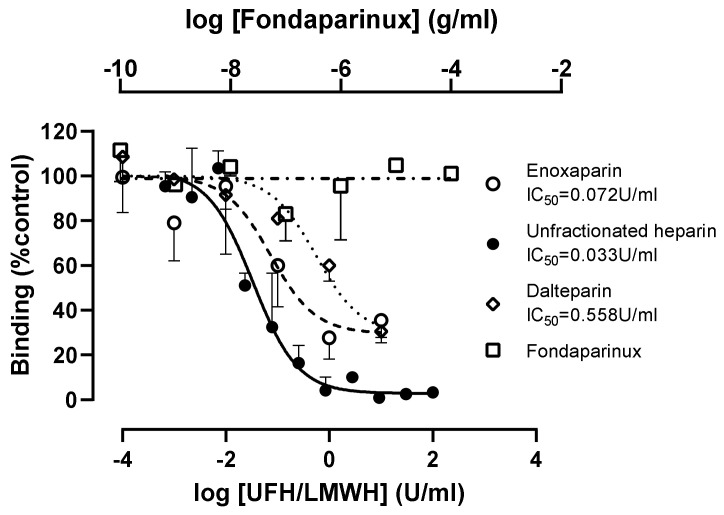
Concentration-dependent inhibition of S1S2 interaction by unfractionated heparin and low-molecular-weight heparins, dalteparin and enoxaparin but not synthetic pentasaccharide, fondaparinux. RT4 cells were pre-incubated with the stated concentrations of unfractionated heparin, enoxaparin, dalteparin and fondaparinux for 30 min at 37 °C, then with 100 nM wtS1S2 for a further 60 min at 37 °C before fluorescent secondary anti-His6 was added for a further 30 min at 21 °C. Cell-associated fluorescence was measured by flow cytometry and is shown as a percentage of the wtS1S2 attachment to untreated control cells. Data are the means ± SEM of 2–3 experiments performed in duplicate.

**Figure 8 cells-10-01419-f008:**
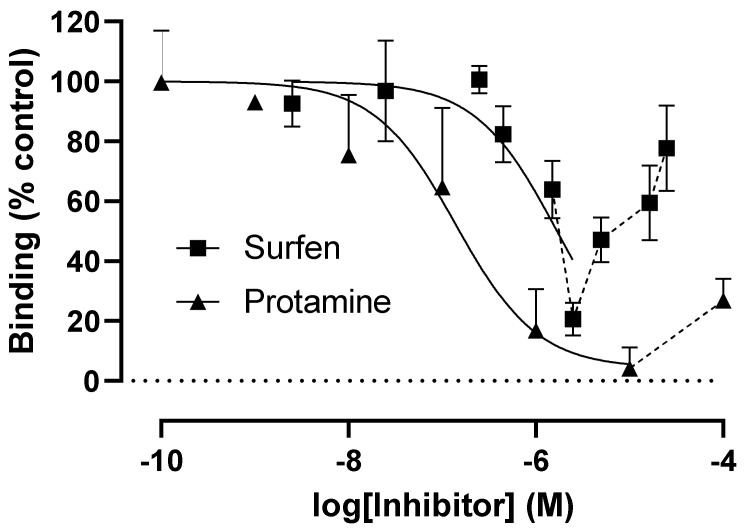
Glycosaminoglycan-binding molecules surfen and protamine sulphate can inhibit wtS1S2 binding. RT4 cells were preincubated with the stated concentrations of surfen or protamine sulphate for 30 min at 37 °C, then with 100 nM wtS1S2 for a further 60 min at 37 °C. Bound wtS1S2 was detected using fluorescent secondary anti-His6, measured by flow cytometry. Data are shown as a percentage of the wtS1S2 attachment to untreated control cells. Data are the means ± SEM of 3–4 experiments performed in duplicate.

**Figure 9 cells-10-01419-f009:**
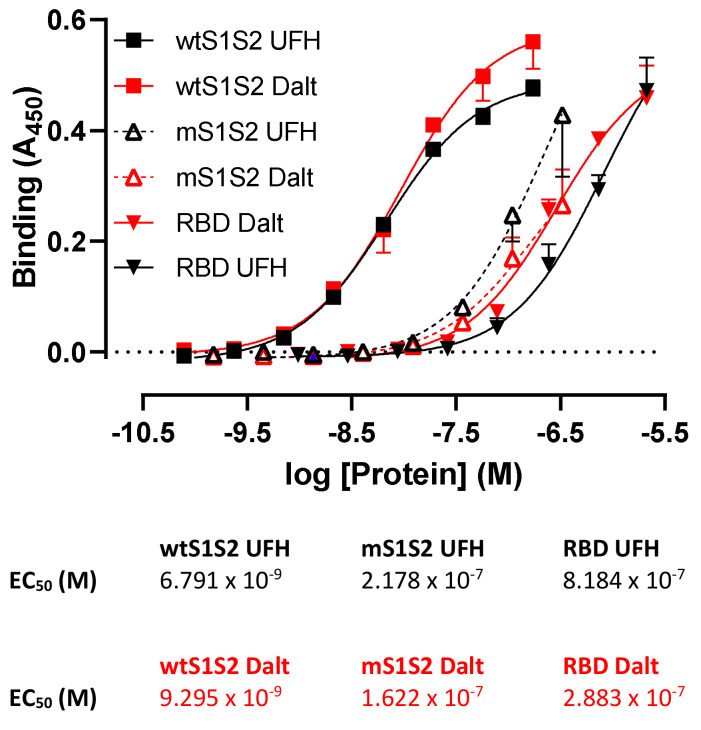
Spike protein binding to heparin requires a polybasic furin cleavage site. Unfractionated heparin (UFH) and low-molecular-weight heparin, dalteparin (Dalt) were immobilised on 96-well plates and used to detect the binding of wtS1S2, mS1S2 lacking the furin cleavage site (mS1S2) or RBD using a biotinylated anti-His6 antibody and streptavidin-HRP. Data shown are the means ± SD from 2–3 separate experiments performed in duplicate. Tables provide the EC_50_ values for each protein.

**Figure 10 cells-10-01419-f010:**
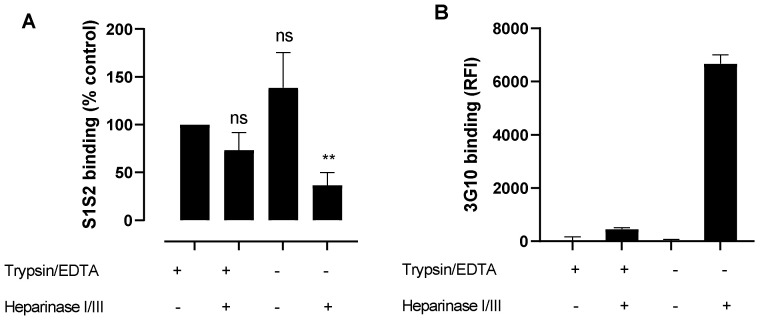
Removal of heparan sulphates only partly inhibits wtS1S2 binding. (**A**) RT4 cells were pretreated for 3 h with heparinase I/III mixture or control growth medium and then harvested by either trypsin/EDTA (+) or non-enzymatic cell dissociation solution (−) and then incubated with 100 nM wtS1S2 at 37 °C before staining with anti-His6 secondary antibody labelled with Dylight 488. The data are shown relative to the non-trypsinised- and non-heparinase-treated control and are the means ± SEM from 3–4 independent experiments performed in duplicate. Significance of difference from the control was assessed by a one-sample t test. ns = not significant; ** *p* < 0.001. (**B**) The effects of heparinase treatment was measured using antibody 3G10 that recognises the cleaved stubs of heparan sulphates. RT4 cells were pre-treated for 3 h with heparinase I/III mixture or control growth medium and then harvested by either trypsin/EDTA or non-enzymatic cell dissociation solution and then incubated with 3G10. Bound antibody was detected by an FITC-labelled secondary and cell-associated fluorescence was measured by flow cytometry. Data are the means ± SD from 2 separate experiments performed in duplicate.

**Figure 11 cells-10-01419-f011:**
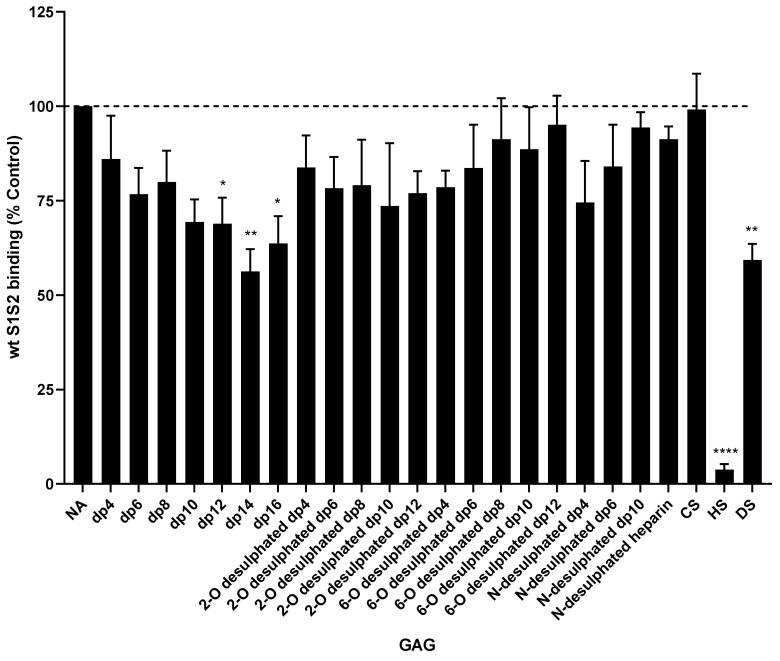
Inhibition of wtS1S2 binding to RT4 cells by heparins and glycosaminoglycans. RT4 cells were incubated with 100 µg/mL of heparin or glycosaminoglycan for 30 min at 37 °C, then with 33 nM wtS1S2 for a further 60 min at 37 °C and bound protein detected using fluorescent secondary anti-His6. Cell-associated fluorescence was measured by flow cytometry and data are shown as a percentage of wtS1S2 attachment to untreated control cells. Data are the means ± SEM of 4–6 separate experiments performed in duplicate. Significance of difference from the control was assessed by a one-sample t test: not significant unless otherwise stated; * *p* < 0.05; ** *p* < 0.001; **** *p* < 0.00001.

**Figure 12 cells-10-01419-f012:**
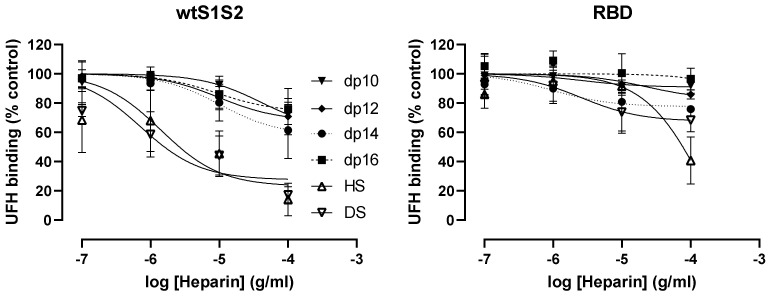
Dose–response curves for the inhibition of wtS1S2 and RBD binding to immobilised unfractionated heparin by selected heparins and glycosaminoglycans. Unfractionated heparin (UFH) was immobilised on 96-well plates and used to detect the binding of wtS1S2, mS1S2 and RBD in the presence of increasing concentrations of heparins of increasing chain length (dp10–16, heparan sulphate (HS) or dermatan sulphate (DS)), detected using a biotinylated anti-His6 antibody and streptavidin-HRP. Data are shown as a percentage of binding to untreated control wells and are the means ± SEM of 4 separate experiments performed in duplicate.

## Data Availability

Not applicable.

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
