# Peer review of "ACE2-Independent Interaction of SARS-CoV-2 Spike Protein with Human Epithelial Cells Is Inhibited by Unfractionated Heparin"

_cells, 2021, doi:10.3390/cells10061419_

Round 1

Reviewer 1 Report

  1. Cite few more published articles showing anti-SARS-COV-2 activity of Heparin (UFH), for example, I would look at this recent article which shows the interaction of Heparin with viral spike protein, also delignate interaction with active sties,although in-silico, it also shows the impact of mutations on their interactions (Gupta et al, Int J Biol Macromol., 2021) and most importantly the anti-viral activity showing it only block the entry, not once virus inside, this can be cited under Results section 3.5 and first para of Discussion.

Author Response

We thank the reviewer for his positive opinion of the manuscript. In response to the comment, we have added the suggested reference to the Introduction (L57) and the Discussion (L257).

Reviewer 2 Report

SARS-CoV-2 binds its cellular surface receptor ACE2 and infects human cells. In addition to ACE2, there are several other receptors for the virus invading have been reported, including NRP1 and glycosaminoglycans. The authors investigated the effects of various heparins on ACE2-independent S protein interaction with cells. They suggested the furin cleavage site on S protein might be a heparin binding site. The manuscript is well written. There is only one major point and some minor points to be addressed for this work.

  1. The authors indicated that the wtS1S2 interactions may require a putative furin binding site. SARS-CoV-1 S protein bound to the cells more weakly than SARS-CoV-2 S that possesses a furin cleavage site. The mechanism behind this statement should be investigated in depth. Is the furin cleavage site sufficient for the binding of the protein to heparin? Short peptides or proteins containing such furin cleavage sequence can be designed and generated. Then the binding ability to cells of such peptides or proteins can be experimentally measured and compared with the S protein of SARS-CoV-1/2 and other controls.

Minor points:

  1. The unit “U/ml” should be given in mg/ml or mg/ml.
  2. “S1S2” should be changed to S protein or spike protein.
  3. Page 15-16, “Receptor binding by the spike protein is dependent on specific conformational changes. …. … mS1S2 naturally hides the RBD domain and requires conformational change to interact with ACE2 on cells.” This discussion text is irrelevant with the major topic and can be removed.

Author Response

We thank the reviewer for his positive opinion of the manuscript and for the excellent suggested experiment. Unfortunately, we are not in a position to perform any further work on this project with no funding or personnel able to work on it. We hope that the reviewer feels that we have sufficient new data to justify publication. 

In response to the minor points:

We cannot convert the U/ml to mg/ml because we used clinical grade heparins that are supplied only with activity data and not the concentration as mg/ml of the active ingredient. It would be possible to estimate the concetration as mg/ml but this would be difficult to translate back to the U/ml concentrations required for comparison to clinically relevant doses.

We use the term 'S1S2' to distinguish this protein from the S1 subunit alone. In response to the reviewer, we have changed the Abstract to mention only the 'spike protein' rather than S1S2 and more carefully define what we mean with the term 'S1S2' in section 2.4 of the Materials and Methods.

We have now deleted the reference to the conformational change in the S1S2 spike protein.

Reviewer 3 Report

The authors describe a possible interaction of spike protein independent of ACE. The article is well written . The authors describe ACE binding domain- Spike protein S1 and S2 (linked by S1 and S2 are linked by a sequence that has a putative furin cleavage site that promotes the infection of human cells)  and arginine amino acid is important part of furin site. However, there are also other interactions. The authors also describe the temperature dependence. However, The authors should motivate the result with more experiments or clarify the questions

Author Response

We thank the reviewer for their careful reading of our manuscript and the many useful suggestions.

Major 1: In response, we would point out that a 1 hour incubation at 4C is sufficient for other binding events measured by the authors (e.g. antibody/antigen, C5a/C5aR1, IgG to Fc receptors). Regarding paraformaldehyde fixation, we are not in a position to perform any further work on this project with no funding or personnel able to work on it. We hope that the reviewer feels that we have sufficient new data to justify publication.

Major 2: We did in fact use an irrelavant protein as a negative control. A His6 tagged form of human complement fragment 5a des Arg was used in early experiments. No binding was detected. This is now mentioned in the Materials and Methods (L78) and Results (L201). We would point out that the His6 tagged RBD also acted as a negative control forS1S2 binding.

Major 3: Although ACE2 mRNA was readily detected in RT4 cells, anti-ACE2 antibody binding was very low, suggesting only low levels at the cell surface. This would make measuring changes in the availability of ACE2 in the presence of S1S2 technically difficult. The lack of inhibition of S1S2 binding by anti-ACE2 antibody strongly supports our contention of ACE2-independent binding.

Minor: No changes in propidium iodide uptake by cells were noted in flow cytometry when S1S2 protein was present. This leads us to believe that spike proteins are not toxic to cells. We have considered combining figures but as our major claim is that the small variations in the low level of ACE2 mRNA expression are not related to differences in S1S2 binding, we do not consider it essential to have the expression data in Fig 1 alongside the binding data in Fig 3.

Round 2

Reviewer 3 Report

The authors have answered all the questions raised except for a minor comment. I understand the authors do not have time to perform the experiment regarding temperature dependence after cell fixation for longer duration at 4 degrees, but this has to be mentioned as a limitation in the discussion section.

The authors, can mention a small note about the limitation of this in the discussion " unlike receptor/ligand and antigen/antibody affinity" our novel finding is more dependent on temperature. However, a long duration at 4 degrees might increase the binding, but with a risk of losing cell viability

Author Response

We thnk the reviewer for their understanding. We have added two sentences to the Conclusions: "A limitation of our study is that we have not explored the mechanism of this temperature-dependency, which might be different to conventional ligand/receptor or antigen/antibody binding. Further experimentation would be required to elucidate the mechanism and to rule out the possibility that longer incubations at lower temperatures would increase S1S2 association with cells."

We hope that this will be seen as acceptable.